# The association of innate and adaptive immunity, subclinical atherosclerosis, and cardiovascular disease in the Rotterdam Study: A prospective cohort study

**Lana Fani**[1], **Kimberly D. van der Willik**[1,2], **Daniel Bos**[1,3], **Maarten J. G. Leening**[1,4], **Peter J. Koudstaal**[5], **Dimitris Rizopoulos**[6], **Rikje Ruiter**[1], **Bruno H. C. Stricker**[1], **Maryam Kavousi**[1], **M. Arfan Ikram**[1], **M. Kamran Ikram**[1,5]*

**1** Department of Epidemiology, Erasmus MC, Rotterdam, the Netherlands, **2** Department of Psychosocial Research and Epidemiology, Netherlands Cancer Institute, Amsterdam, the Netherlands, **3** Department of Radiology and Nuclear Medicine, Erasmus MC, Rotterdam, the Netherlands, **4** Department of Cardiology, Erasmus MC, Rotterdam, the Netherlands, **5** Department of Neurology, Erasmus MC, Rotterdam, the Netherlands, **6** Department of Biostatistics, Erasmus MC, Rotterdam, the Netherlands

* m.ikram@erasmusmc.nl

**Data Availability Statement:** Rotterdam Study data can be made available to interested researchers upon request. Requests can be

## Abstract

### Background

Atherosclerotic cardiovascular disease (ASCVD) is driven by multifaceted contributions of the immune system. However, the dysregulation of immune cells that leads to ASCVD is poorly understood. We determined the association of components of innate and adaptive immunity longitudinally with ASCVD, and assessed whether arterial calcifications play a role in this association.

### Methods and findings

Granulocyte (innate immunity) and lymphocyte (adaptive immunity) counts were determined 3 times (2002–2008, mean age 65.2 years; 2009–2013, mean age 69.0 years; and 2014–2015, mean age 78.5 years) in participants of the population-based Rotterdam Study without ASCVD at baseline. Participants were followed-up for ASCVD or death until 1 January 2015. A random sample of 2,366 underwent computed tomography at baseline to quantify arterial calcification volume in 4 vessel beds. We studied the association between immunity components with risk of ASCVD and assessed whether immunity components were related to arterial calcifications at baseline. Of 7,730 participants (59.4% women), 801 developed ASCVD during a median follow-up of 8.1 years. Having an increased granulocyte count increased ASCVD risk (adjusted hazard ratio for doubled granulocyte count [95% CI] = 1.78 [1.34–2.37], P < 0.001). Higher granulocyte counts were related to larger calcification volumes in all vessels, most prominently in the coronary arteries (mean difference in calcium volume [mm$^3$] per SD increase in granulocyte count [95% CI] = 32.3 [9.9–54.7], P < 0.001). Respectively, the association between granulocyte count and incident coronary heart disease and stroke was partly mediated by coronary artery calcification (overall proportion

directed to data manager Frank J.A. van Rooij (f.
vanrooij@erasmusmc.nl).

**Funding:** This work was supported by the
European Union's Horizon 2020 research and
innovation programme [grant number 667375]
("CoSTREAM"); the Dutch Cancer Society [grant
number NKI-20157737]; the Erasmus Medical
Center and Erasmus University Rotterdam; the
Netherlands Organization for Scientific Research
(NWO) [grant numbers 948-00-010, 918-46-615];
the Netherlands Organization for Health Research
and Development (ZonMw); the Research Institute
for Diseases in the Elderly (RIDE); the Ministry of
Education, Culture and Science; the Ministry of
Health, Welfare and Sports; the European
Commission (DG XII); and the Municipality of
Rotterdam. Maryam Kavousi is supported by the
VENI grant (91616079) from ZonMw. The funding
source had no role in study design, collection,
analysis, interpretation of data, writing of the report
or decision to submit the article for publication.

**Competing interests:** The authors have declared
that no competing interests exist.

**Abbreviations:** ASCVD, atherosclerotic
cardiovascular disease; CHD, coronary heart
disease; CT, computed tomography; GLR,
granulocyte-to-lymphocyte ratio; HDL, high-density
lipoprotein; HR, hazard ratio; hs-CRP, high-
sensitivity C-reactive protein; MI, myocardial
infarction; NSAID, nonsteroidal anti-inflammatory
drug; PLR, platelet-to-lymphocyte ratio; SII,
systemic immune-inflammation index.

mediated [95% CI] = 19.0% [−10% to 32.3%], $P = 0.08$) and intracranial artery calcification (14.9% [−10.9% to 19.1%], $P = 0.05$). A limitation of our study is that studying the etiology of ASCVD remains difficult within an epidemiological setting due to the limited availability of surrogates for innate and especially adaptive immunity.

## Conclusions

In this study, we found that an increased granulocyte count was associated with a higher risk of ASCVD in the general population. Moreover, higher levels of granulocytes were associated with larger volumes of arterial calcification. Arterial calcifications may explain a proportion of the link between granulocytes and ASCVD.

## Author summary

### Why was this study done?

- Many people develop cardiovascular disease during their lifetime.
- Recent findings suggest a role of the immune system in the pathogenesis of cardiovascular disease.
- Currently, there is no low-cost treatment identified to treat dysregulations of the immune system for preventing cardiovascular disease, because the exact immune cells involved in cardiovascular pathology are unknown.

### What did the researchers do and find?

- We repeatedly quantified the number of innate immune cells (granulocytes and platelets) and adaptive immune cells (lymphocytes) from 7,730 people who participated in the Dutch population-based Rotterdam Study and determined the relation of these cells with the risk of cardiovascular disease.
- The risk of cardiovascular disease was higher among people with increased granulocytes over time, while the risk was lower among people with increased lymphocytes.

### What do these findings mean?

- The findings suggest that higher activity of the innate immune system and lower activity of the adaptive immune system may be associated with higher risk of cardiovascular disease.
- The relation between granulocytes and the risk of cardiovascular disease may be mediated by subclinical atherosclerosis.
- The findings warrant future studies that could focus more specifically on targeting both innate and adaptive immunity, rather than only innate immunity, and refine insights into these pathways.

## Introduction

Atherosclerotic cardiovascular disease (ASCVD) arises from various interacting pathophysiological processes [1]. Recent findings point towards a key role of the immune system, which can be broadly classified into innate and adaptive immunity [2]. Innate immunity refers to immune responses present at birth, whereas adaptive immunity is acquired during life by exposure to antigens [3]. The role of innate immunity in the pathophysiology of ASCVD has been recognized on the basis of evidence from experimental and observational data [4–8]. To date, however, no easy, accessible, low-cost treatment has been identified to effectively target the innate immune system for preventing ASCVD [8]. Given the mechanistic diversity of inflammatory pathways, considering other pathways could widen avenues for therapeutic targets, for example targeting the adaptive immune system. Adaptive immune cells may provide protective responses at atherosclerotic sites, as has been shown in neurodegenerative disorders [9,10] and also in cardiovascular disease in experimental and clinical studies [11]. Moreover, repeatedly measuring inflammation may better capture the biologically dynamic processes of these 2 immune systems than a single measurement, because of the assessment of changes over time.

Recent work from the field of cancer research suggests that measuring granulocytes and platelets provides important markers of innate immunity [12–14], whereas measuring lymphocytes yields information on adaptive immunity [15]. Furthermore, combining these measurements into ratios, i.e., the granulocyte-to-lymphocyte ratio (GLR), platelet-to-lymphocyte ratio (PLR), and systemic immune-inflammation index (SII), is thought to even better reflect the relative balance between innate and adaptive immunity [16–18]. In this study, we first determined the longitudinal association of these immunity components with risk of ASCVD in the general population by using the framework of joint models for longitudinal and survival data, and hypothesized that an increase in innate immunity markers and decrease in adaptive immunity lead to increased risk of ASCVD. Second, we examined the association between the immunity components and arterial calcifications to explore atherosclerosis as a possible mediator.

## Methods

### Study population

The present study is embedded within the Rotterdam Study, a prospective population-based cohort study in Rotterdam, the Netherlands. The Rotterdam Study started in 1989 with 7,983 persons (78% response rate) aged ≥55 years and residing in the district Ommoord, a suburb of Rotterdam. This first subcohort (RS-I) was extended with a second subcohort (RS-II) in 2000, consisting of 3,011 persons (67% response rate) aged ≥55 years, and with a third subcohort (RS-III) in 2006, composed of 3,932 persons aged ≥45 years (65% response rate). The design of the Rotterdam Study has been described in detail previously [19]. In brief, participants were examined at study entry and at follow-up visits every 3 to 5 years. They were interviewed at home by a trained research nurse, followed by 2 visits at the research facility for additional interviewing, laboratory assessments, and imaging.

The Rotterdam Study has been approved by the Medical Ethics Committee of the Erasmus MC (registration number MEC 02.1015) and by the Dutch Ministry of Health, Welfare and Sport (Population Screening Act [WBO], license number 1071272-159521-PG). The Rotterdam Study personal registration data collection is filed with the Erasmus MC Data Protection Officer under registration number EMC1712001. The Rotterdam Study has been entered into the Netherlands Trial Register (https://www.trialregister.nl) and into the WHO International

Clinical Trials Registry Platform (https://www.who.int/ictrp/network/primary/en/) under the shared catalogue number NTR6831. All participants provided written informed consent to participate in the study and to have their information obtained from treating physicians.

For the current study, the analysis plan was drafted in January 2019 (S1 Analysis Plan). This study is reported as per the Strengthening the Reporting of Observational Studies in Epidemiology (STROBE) guideline (S1 STROBE Checklist).

Laboratory tests for granulocytes, platelets, and lymphocytes were introduced from 2002 onwards, corresponding with the following assessment rounds in the Rotterdam Study (baseline in this study): fourth round of RS-I, second round of RS-II, and first round of RS-III (*n* = 9,996 in total).

## Assessment of blood cell counts and their derived ratios

Fasting blood samples were drawn during each visit at the research center, with a maximum of 3 visits during follow-up. From 7,730 participants, 5,085 participants were available for a second blood cell assessment, and 270 participants were available for a third blood cell assessment. Full blood count measurements were performed using the Coulter AcT diff2 Hematology Analyzer (Beckman Coulter, Brea, CA, US) directly after the blood sample was drawn. Laboratory measurements included absolute granulocyte, platelet, and lymphocyte counts in $10^9$ per liter. GLR and PLR were calculated as the ratio of granulocyte count to lymphocyte count and as the ratio of platelet count to lymphocyte count, respectively. SII was defined as platelet count times GLR [20].

## Assessment of stroke and coronary heart disease

The clinical outcomes of interest included first fatal or nonfatal stroke or coronary heart disease (CHD) separately as well as combined into ASCVD as previously described. Follow-up data were collected through general practitioners and subsequent collection of information from letters by medical specialists and discharge reports, in cases of hospitalization. CHD included (1) nonfatal myocardial infarction (MI) and (2) CHD mortality (mortality with definite MI, definite CHD, or possible CHD as underlying cause of death) [21]. Stroke was defined as a syndrome of rapidly developing symptoms of focal or global cerebral dysfunction lasting ≥24 hours or leading to death, with apparent vascular cause [22,23]. We categorized strokes as ischemic or hemorrhagic, based on neuroimaging reports and clinical symptoms, or as unspecified if we were unable to differentiate the stroke type further. Subarachnoid hemorrhages due to ruptured aneurysms were not considered as stroke. Follow-up was virtually complete until 1 January 2015 (96.6% of potential person-years observed). ASCVD furthermore included other ASCVD mortality.

## Assessment of arterial calcification

Non-contrast computed tomography (CT) images were obtained using a 16-slice or 64-slice multidetector CT scanner (Somatom Sensation 16 or 64; Siemens, Forchheim, Germany). Using a cardiac scan and an extracardiac scan that reached from the aortic root to the intracranial vasculature (1 centimeter above the sella turcica), we visualized the following vessels: coronary arteries, aortic arch, extracranial internal carotid arteries, and intracranial internal carotid arteries. Radiation dose and other imaging parameters of both scans are described elsewhere [24,25].

The amount of calcification in the coronary arteries, aortic arch, and extracranial carotid arteries was quantified using widely used, commercially available dedicated software (Syngo.via; Siemens). Calcification in the coronary arteries comprised a summation of calcification in

the left main, left anterior descending, left circumflex, and right coronary artery. Calcification in the aortic arch was scored from the origin of the aortic arch (defined as the image in which the ascending and descending aorta merge into the inner curvature of the aortic arch) to the first 1 cm of the branches originating from the arch. Calcification in the extracranial carotid arteries was assessed within 3 cm proximal and distal of the bifurcation on both sides and summed.

Intracranial internal carotid artery calcification was assessed from the horizontal segment of the petrous internal carotid artery to the top of the internal carotid artery. Scoring was done semi-automatically, because automated software is not yet available for this region (the close relationship between calcification and the skull base precludes the use of commercial software given a high false-positive rate). Regions of interest were manually drawn in the course of the intracranial internal carotid artery on both sides, after which the number of pixels within these regions above 130 Hounsfield units was summed and multiplied by the pixel size and the slice increment in order to obtain a volume in cubic millimeters [26,27].

## Covariates

We assessed education, smoking, and use of antihypertensive, lipid-lowering, and antidiabetic medication at baseline by interview. Diastolic and systolic blood pressures were measured twice on the right arm with a random-zero sphygmomanometer, of which the mean was used. Total cholesterol and high-density lipoprotein (HDL) cholesterol were measured with a Coulter AcT diff2 Hematology Analyzer. Body mass index (BMI) was computed from measurements of height and weight ($kg/m^2$). Diabetes mellitus was defined as use of antidiabetic medication, fasting serum glucose level $\geq$ 7.1 mmol/l ($\geq$127.9 mg/dl), or random non-fasting serum glucose level $\geq$ 11.1 mmol/l ($\geq$200.0 mg/dl) [28]. High-sensitivity C-reactive protein (hs-CRP) was determined in serum that was drawn during the third assessment round in the Rotterdam Study, i.e., third round of RS-I (1996–1999), which was stored at −20˚C until performance of the hs-CRP measurements in 2003 to 2004. hs-CRP was measured using Rate Near Infrared Particle Immunoassay (Immage Immunochemistry System, Beckman Coulter). History of cancer was obtained from general practitioners' medical records (including hospital discharge letters), the Dutch Hospital Data registry, and regional histopathology and cytopathology registries. Furthermore, we assessed history of immune-modulating medication use— nonsteroidal anti-inflammatory drugs (NSAIDs) (ATC code M01), immunosuppressives (ATC code L04), and methotrexate (ATC code L01BA01)—by number of prescriptions between 1 January 1995 and blood draw date.

## Statistical analysis

Because the blood cell counts and their derived ratios have a skewed distribution and to adhere to the linearity assumptions, the analysis was based on the natural logarithmic (Ln) transformation of their values. We first determined the association of the granulocyte, platelet, and lymphocyte counts and their derived ratios with the risk of stroke and CHD separately as well as ASCVD, using the framework of joint models for longitudinal and survival data [29]. For the longitudinal data, a linear mixed effects model was used, in which we specified the immunity component as the dependent variable and time as the independent variable to assess the mean change in immunity components. Random intercepts and slopes were used to incorporate individual response trajectories. For modeling survival data, we used a Cox regression model in which we included the true underlying profile of the blood cell counts as estimated from the longitudinal model. Here the baseline risk function assumed a piecewise constant with 6 knots placed at equally spaced percentiles of the observed event times. Joint models

allow accounting for (1) measurement error during follow-up, i.e., biological variation, (2) the effect of factors at an earlier time point on the values of blood cell measurements at a later time point [30], and (3) the correlations in the repeated measurements of the blood cell counts. Because we are using log-log models ($\log Y_i = \alpha + \beta \log X_i + \varepsilon$), we report hazard ratios (HRs) with 95% confidence intervals (CIs) obtained by exponentiating Ln(2) times the estimated coefficients from the Cox model. These HRs correspond to the increase in risk for a doubling of the blood cell count or ratio. We computed 2 nested models: Model I was adjusted for baseline age (continuous, centered as age minus mean age) and sex; model II was additionally adjusted for education, smoking status, BMI, diabetes mellitus, systolic and diastolic blood pressure (continuous, centered as blood pressure minus mean blood pressure), antihypertensive medication, total cholesterol, HDL cholesterol, and lipid-lowering medication. For assessment of the association between individual blood cell counts and ASCVD, we performed the analyses with adjustment for the baseline blood cell counts of the remaining 2 blood cell types. Follow-up time was used as the time scale and started at the date of first laboratory assessment; follow-up continued until date of stroke, CHD, death, loss to follow-up, or 1 January 2015, whichever came first. We also investigated the association of the blood cell counts and their ratios with ischemic and hemorrhagic stroke separately. We assessed the joint model assumptions by examining the proportional hazards assumption using Schoenfeld residuals, and goodness of fit using Cox–Snell residuals for the survival part and marginal residuals for the longitudinal part.

In sensitivity analyses, we additionally excluded participants with a history of cancer or a history of immune-modulating medication at baseline. In addition, we performed additional adjustments for NSAID use. We explored effect modification by stratifying by age, sex, hs-CRP (at a cutoff of 2 mg/l), use of lipid-lowering medication, coronary artery calcification volume (at a cutoff of 10 mm$^3$), and smoking, all at baseline. We formally tested interaction between these factors and blood cell counts on the multiplicative scale by adding interaction terms to model II. Lastly, we repeated the analyses for the granulocytes and lymphocytes and risk of ASCVD using age as the time scale instead of follow-up time to account for potential residual confounding by age and to minimize potential effects of left truncation.

Next, in order to explore possible mechanisms, we performed a cross-sectional analysis assessing the association of the granulocyte, platelet, and lymphocyte counts and their derived ratios at baseline, per standard deviation increase, with calcification volume in each vessel bed (coronary arteries, aortic arch, extracranial carotid arteries, and intracranial carotid arteries) using linear regression models. To facilitate interpretation of these analyses, we chose to first standardize the blood cell counts only, to make results comparable, and keep the rest of the variables untransformed. We adjusted these analyses for age and sex (model I), and additionally for education, smoking status, BMI, diabetes mellitus, systolic blood pressure, diastolic blood pressure, antihypertensive medication, total cholesterol, HDL cholesterol, and lipid-lowering medication (model II). To adhere to the homoscedasticity and linearity assumptions, we repeated linear regression models after natural log-transforming exposure and outcome variables. As a sensitivity analysis, we looked at the association between the granulocyte, platelet, and lymphocyte counts and their derived ratios and continuous arterial calcification only in those with non-zero arterial calcification, since the magnitude of the increase from 0 to 1 for volume may not be the same as other incremental increases in arterial calcification.

Finally, we performed a mediation analysis to assess a possible mediating effect of calcification volume in the coronary arteries and intracranial carotid arteries on the association between innate immunity and the risk of CHD and stroke, respectively. We chose the innate immunity marker displaying the strongest association with both calcifications and incident CHD or stroke for this analysis. We tested the overall proportion mediated by calcifications.

To test for mediation effects, we used the decomposition analysis techniques as described by VanderWeele [31]. Participants with a history of CHD or stroke were excluded from the analysis. We analyzed the association of innate immunity with calcifications with linear regression models. The association of innate immunity or calcifications with CHD or stroke was investigated by Cox regression models.

Multiple imputation by chained equations was used for missing covariates (maximum of 1.5%), with 5 imputed datasets based on other covariates and the outcome. Rubin's method was used for pooling the 5 analyses to obtain pooled HRs and 95% CIs [32]. Two-sided $P < 0.05$ was considered statistically significant for all analyses. Statistical analyses were performed using the R packages *mice*, *survival*, *nlme*, *JM*, and *JMbayes* in RStudio Version 3.3.2 [29,30,33,34].

## Results

From 8,712 participants, we excluded those with a history of ASCVD ($n$ = 819) at baseline, those who had incomplete data regarding their history of ASCVD ($n$ = 95), and those who did not provide informed consent to assess medical records during follow-up ($n$ = 60). Lastly, we excluded participants with incomplete laboratory assessment ($n$ = 8), resulting in 7,730 participants for analysis (flowchart in Fig 1). From the fourth round of RS-I and the second round of RS-II, a random sample was invited to undergo non-contrast CT to quantify arterial calcification. Of these participants, 2,366 had complete calcification and blood assessment.

Characteristics of the study participants are presented in Table 1. Mean age of included study participants was 65.2 years, and 59.4% were women. During a median (interquartile range) follow-up of 8.1 (3.4) years (62,095 person-years), 801 participants developed ASCVD, of whom 423 participants developed CHD and 378 stroke. ASCVD-free survival probability and average change of granulocytes and lymphocytes over time are presented in S1 Fig in the total population and in S2 Fig in younger and older individuals separately.

An increased granulocyte count, but not platelet count, was associated with an increased risk of ASCVD (adjusted HR [95% CI] for doubled granulocyte count = 1.78 [1.34–2.37], $P <$ 0.001, and for doubled platelet count = 1.17 [0.90–1.51], $P$ = 0.24, Table 2). With respect to

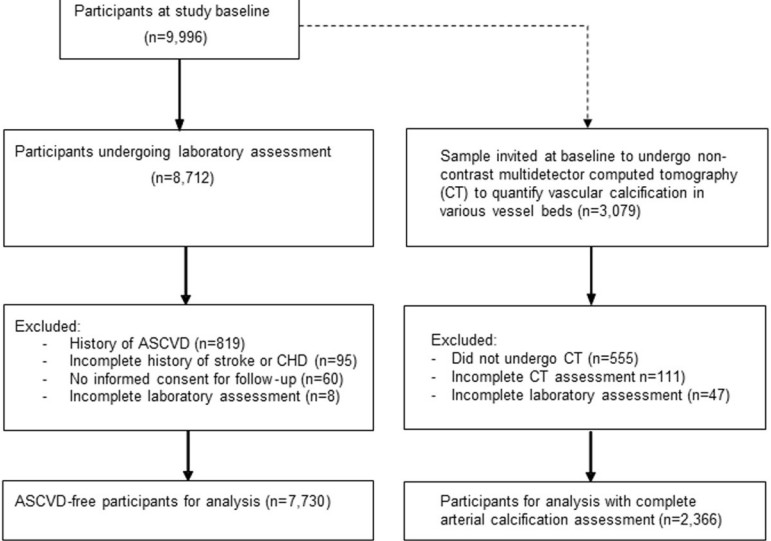

**Fig 1. Flowchart of study population.** ASCVD, atherosclerotic cardiovascular disease; CHD, coronary heart disease.

**Table 1. Baseline characteristics.**

| Characteristic | Laboratory assessment cohort N = 7,730 | Sample undergoing CT N = 2,366 |
|---|---|---|
| Women | 4,594 (59.4%) | 1,236 (52.2%) |
| Age, years | 65.2 (10.2) | 69.1 (6.7) |
| Smoking | | |
| Ever | 5,278 (68.7%) | 1,662 (71.2%) |
| Never | 2,401 (31.3%) | 670 (28.7%) |
| Diabetes mellitus | 415 (5.4%) | 129 (5.5%) |
| BMI, kg/m$^2$ | 27.6 (4.4) | 27.6 (4.0) |
| Education | | |
| Primary | 830 (10.8%) | 179 (7.7%) |
| Lower | 3,124 (40.7%) | 979 (42.1%) |
| Intermediate | 2,205 (28.8%) | 716 (30.8%) |
| Higher | 1,509 (19.7%) | 451 (19.4%) |
| Systolic blood pressure, mm Hg | 142.3 (21.8) | 146.7 (20.0) |
| Diastolic blood pressure, mm Hg | 81.3 (11.0) | 80.2 (10.7) |
| Antihypertensive drugs | 2,571 (33.6%) | 921 (39.5%) |
| Total cholesterol, mmol/l | 5.7 (1.0) | 5.6 (1.0) |
| HDL cholesterol, mmol/l | 1.5 (0.4) | 1.4 (0.4) |
| Lipid-lowering medication | 1,429 (18.7%) | 535 (23.0%) |
| Granulocyte count, ×10$^3$/μl | 4.0 (1.4) | 4.0 (1.3) |
| Platelet count, ×10$^3$/μl | 270.9 (66.6) | 259.0 (64.0) |
| Lymphocyte count, ×10$^3$/μl | 2.3 (1.1) | 2.3 (1.0) |
| GLR | 1.9 (0.8) | 1.9 (0.9) |
| PLR | 127.9 (46.4) | 124.9 (45.8) |
| SII | 509.3 (270.7) | 494.3 (272.2) |

Data presented as mean (standard deviation) for continuous variables and number (percentage) for categorical variables. Number of missing values for the ASCVD-free cohort: 51 (0.7%) for smoking, 22 (0.3%) for diabetes, 25 (0.3%) for BMI, 62 (0.8%) for education, 41 for systolic and diastolic blood pressure (0.5%), 69 for blood pressure lowering medication (0.9%), 25 for cholesterol (0.3%), 27 for HDL cholesterol (0.3%), and 69 for lipid lowering medication (0.9%). Number of missing values for the sample undergoing CT are: 52 (0.6%) for smoking, 22 (0.3%) for diabetes, 26 (0.3%) for BMI, 41 (0.5%) for education, 14 for systolic and diastolic blood pressure (0.6%), 35 for blood pressure lowering medication (1.5%), 6 for cholesterol (0.3%), 6 for HDL cholesterol (0.3%) and 35 for lipid lowering medication (1.5%).

BMI, body mass index; CT, computed tomography; GLR, granulocyte-to-lymphocyte ratio; PLR, platelet-to-lymphocyte ratio; SII, systemic immune-inflammation index.

adaptive immunity, an increased level of lymphocytes was associated with a decreased risk of ASCVD, although this result was not statistically significant (adjusted HR for doubled lymphocyte count [95% CI] = 0.87 [0.71–1.06], $P$ = 0.17). Both an increased GLR and an increased SII were associated with increased ASCVD risk (adjusted HR for doubled GLR [95% CI] = 1. 37 [1.14–1.65], $P$ = 0.001, and for doubled SII = 1.19 [1.03–1.36], $P$ = 0.02), whereas an increased PLR was not associated with ASCVD risk. We observed no major violations of the joint model assumptions. Effect estimates did not materially change, and in fact became stronger, when we excluded participants with a history of cancer or history of immune-modulating medication at baseline (723 ASCVD cases among 7,063 remaining participants) (adjusted HR for doubled granulocyte count [95% CI] = 1.90 [1.41–2.57], $P$ < 0.001, and for doubled lymphocyte count = 0.82 [0.65–1.04], $P$ = 0.10). Additional adjustment for NSAID use did not change the effect estimates. We found that the association between higher levels of granulocytes and risk of ASCVD was more pronounced in participants aged younger than the median age of 64

**Table 2. Joint models for the association between repeated blood-based immunity markers and subsequent risk of ASCVD.**

| Model and immunity marker | ASCVD n/N = 801/7,730 | | Stroke n/N = 378/7,730 | | CHD n/N = 423/7,730 | |
|---|---|---|---|---|---|---|
| | HR (95% CI) | P value | HR (95% CI) | P value | HR (95% CI) | P value |
| **Model I** | | | | | | |
| *Innate immunity** | | | | | | |
| Granulocytes | 2.63 (2.03–3.40) | <0.001 | 2.14 (1.48–3.11) | <0.001 | 3.10 (2.18–4.42) | <0.001 |
| Platelets | 1.10 (0.85–1.41) | 0.47 | 1.27 (0.88–1.83) | 0.21 | 0.96 (0.68–1.36) | 0.83 |
| *Adaptive immunity** | | | | | | |
| Lymphocytes | 1.03 (0.85–1.25) | 0.77 | 0.96 (0.72–1.29) | 0.80 | 1.09 (0.83–1.43) | 0.54 |
| *Balance between innate and adaptive immunity* | | | | | | |
| GLR | 1.52 (1.27–1.82) | <0.001 | 1.47 (1.13–1.91) | 0.004 | 1.56 (1.22–2.01) | <0.001 |
| PLR | 1.02 (0.87–1.19) | 0.84 | 1.15 (0.91–1.46) | 0.24 | 0.97 (0.78–1.20) | 0.76 |
| SII | 1.25 (1.09–1.43) | 0.002 | 1.27 (1.04–1.56) | 0.02 | 1.24 (1.03–1.51) | 0.03 |
| **Model II** | | | | | | |
| *Innate immunity** | | | | | | |
| Granulocytes | 1.78 (1.34–2.37) | <0.001 | 1.50 (1.00–2.27) | 0.05 | 1.98 (1.34–2.93) | 0.001 |
| Platelets | 1.17 (0.90–1.51) | 0.24 | 1.38 (0.95–2.01) | 0.09 | 1.01 (0.71–1.44) | 0.95 |
| *Adaptive immunity** | | | | | | |
| Lymphocytes | 0.87 (0.71–1.06) | 0.17 | 0.85 (0.63–1.15) | 0.29 | 0.88 (0.66–1.17) | 0.37 |
| *Balance between innate and adaptive immunity* | | | | | | |
| GLR | 1.37 (1.14–1.65) | 0.001 | 1.31 (1.00–1.70) | 0.05 | 1.37 (1.07–1.78) | 0.01 |
| PLR | 1.17 (0.99–1.38) | 0.06 | 1.25 (0.98–1.59) | 0.07 | 1.10 (0.88–1.38) | 0.41 |
| SII | 1.19 (1.03–1.36) | 0.02 | 1.21 (0.99–1.48) | 0.06 | 1.16 (0.96–1.41) | 0.13 |

HR is per doubled blood cell count. Model I is adjusted for age and sex. Model II is adjusted for age, sex, education, smoking status, body mass index, diabetes mellitus, systolic and diastolic blood pressure, antihypertensive medication, high-density lipoprotein cholesterol, total cholesterol, and lipid-lowering medication. All markers were logarithmically transformed.

*Analysis for each blood cell type adjusted for the baseline blood cell counts of the remaining 2 blood cell types.

ASCVD, atherosclerotic cardiovascular disease; CHD, coronary heart disease; CI, confidence interval; GLR, granulocyte-to-lymphocyte ratio; HR, hazard ratio; *n*, number of incident events; *N*, number of participants for analysis; PLR, platelet-to-lymphocyte ratio; SII, systemic immune-inflammation index (defined as platelet count times GLR).

years compared to those 64 years or older (adjusted HR for doubled granulocyte count [95% CI] = 2.32 [1.20–4.47], $P < 0.001$) ($P_{interaction}$ = 0.002). Results of all stratified analyses with interaction terms are shown in S1 Table. Lastly, risk estimates were comparable when using age as the time scale instead of follow-up time.

When examining CHD and stroke as separate events, an increased granulocyte count was associated with an increased risk of both events (adjusted HR for doubled granulocyte count [95% CI] = 1.98 [1.34–2.93], $P$ = 0.001, for CHD and 1.50 [1.00–2.27], $P$ = 0.05, for stroke; Table 2). An increased platelet count was associated with higher risk of stroke only (adjusted HR for doubled platelet count [95% CI] = 1.38 [0.95–2.01], $P$ = 0.09). An increased lymphocyte count showed similar protective effects for both CHD and stroke, although these effects were not statistically significant (adjusted HR for doubled lymphocyte count [95% CI] = 0.88 [0.66–1.17], $P$ = 0.37, for CHD and 0.85 [0.63–1.15], $P$ = 0.29, for stroke). With respect to the ratio measures, an increased GLR was associated with both increased CHD and stroke risk (adjusted HR for doubled GLR [95% CI] = 1.37 [1.07–1.78], $P$ = 0.01, for CHD and 1.31 [1.00–1.70], $P$ = 0.05, for stroke). When investigating ischemic and hemorrhagic stroke separately, an increased granulocyte count was associated with an increased risk of ischemic stroke (HR for doubled

**Table 3. Models for the association between blood-based immunity markers and arterial calcification volume.**

| Immunity marker[a] | Arterial calcification volume[b] | | | | | | | |
|---|---|---|---|---|---|---|---|---|
| | Coronary arteries | | Aortic arch | | Extracranial internal carotid | | Intracranial internal carotid | |
| | Mean difference (95% CI) | P value | Mean difference (95% CI) | P value | Mean difference (95% CI) | P value | Mean difference (95% CI) | P value |
| *Innate immunity** | | | | | | | | |
| Granulocytes | 0.26 (0.14; 0.39) | <0.001 | 0.20 (0.07; 0.32) | <0.001 | 0.22 (0.09; 0.35) | <0.001 | 0.18 (0.05; 0.32) | 0.01 |
| Platelets | −0.02 (−0.18; 0.13) | 0.76 | 0.01 (−0.14; 0.17) | 0.87 | −0.05 (−0.21; 0.11) | 0.57 | −0.09 (−0.25; 0.07) | 0.27 |
| *Adaptive immunity** | | | | | | | | |
| Lymphocytes | 0.15 (0.03; 0.27) | 0.02 | 0.01 (−0.11; 0.13) | 0.91 | 0.09 (−0.04; 0.21) | 0.17 | 0.08 (−0.04; 0.21) | 0.19 |
| *Balance between innate and adaptive immunity* | | | | | | | | |
| GLR | 0.06 (−0.04; 0.15) | 0.24 | 0.10 (0.00; 0.19) | 0.04 | 0.06 (−0.03; 0.16) | 0.18 | 0.04 (−0.05; 0.14) | 0.36 |
| PLR | −0.09 (−0.19; 0.01) | 0.08 | 0.01 (−0.09; 0.11) | 0.85 | −0.06 (−0.17; 0.04) | 0.25 | −0.08 (−0.18; 0.03) | 0.15 |
| SII | 0.06 (−0.01; 0.14) | 0.10 | 0.09 (0.01; 0.16) | 0.03 | 0.06 (−0.02; 0.13) | 0.15 | 0.03 (−0.05; 0.11) | 0.48 |

Adjusted for age, sex, education, smoking status, body mass index, diabetes mellitus, systolic and diastolic blood pressure, antihypertensive medication, high-density lipoprotein cholesterol, total cholesterol, and lipid-lowering medication.

[a] All markers were natural log-transformed (Ln[immunity component × $10^3$/μl]).

[b] We added 1.0 mm$^3$ to the non-transformed values to deal with calcification volumes of 0 and used standardized natural log-transformed values (Ln[calcification volume + 1.0 mm$^3$]).

* Analysis for each blood cell type adjusted for the baseline blood cell counts of the remaining 2 blood cell types.

CI, confidence interval; GLR, granulocyte-to-lymphocyte ratio; PLR, platelet-to-lymphocyte ratio; SII, systemic immune-inflammation index.

granulocyte count [95% CI] = 1.44 [0.90–2.31], $P = 0.13$) and hemorrhagic stroke (2.48 [0.86–7.14], $P = 0.09$), whereas platelets only increased risk of ischemic stroke (HR for doubled platelet count [95% CI] = 1.44 [0.93–2.22], $P = 0.10$).

Regarding the association of immunity with arterial calcification, we found that higher levels of granulocytes at baseline related to larger calcification volumes in all arteries, but most prominently in the coronary arteries (mean difference in calcium volume [mm$^3$] per SD increase in granulocyte count [95% CI] = 32.3 [9.1 to 54.7], $P < 0.001$). Regarding adaptive immunity, higher levels of lymphocytes were not related to calcification volumes. Higher levels of GLR and SII were related to larger calcification volumes in the aortic arch only (mean difference in calcium volume [mm$^3$] per SD increase [95% CI] = 48.6 [2.5 to 94.8], $P = 0.04$, and 58.8 [13.8 to103.8], $P = 0.03$, respectively). In Table 3, the linear regression models after transforming the exposure and outcome variables are presented. Our sensitivity analyses with non-zero calcifications similarly showed associations between higher levels of granulocytes and larger calcification volumes in all arteries, but the analysis with extracranial internal carotid artery calcification as outcome did not reach statistical significance.

The association between granulocytes and incident CHD was partly mediated by coronary artery calcifications (overall proportion mediated [95% CI] = 19.0% [−10% to 32.3%], $P = 0.08$), while the association between granulocytes and incident stroke was partly mediated by intracranial artery calcifications (14.9% [−10.9% to 19.1%], $P = 0.05$), albeit this finding was not statistically significant.

## Discussion

In this study, we found that an increase of innate immunity markers over time, as reflected by a doubling of the granulocyte count and the ratios GLR and SII, was robustly associated with a higher risk of ASCVD. Furthermore, higher levels of granulocytes at baseline were related to larger burden of subclinical atherosclerosis, as measured by arterial calcifications. Coronary

artery calcifications mediated 19.0% of the association between granulocytes and CHD, while intracranial artery calcifications mediated 14.9% of the association between granulocytes and stroke.

To date, few population-based studies have investigated the association between innate immunity markers and cardiovascular outcomes. Elevated GLR has been linked to ischemic stroke incidence and cardiovascular disease [35,36]. A meta-analysis of 38 studies showed that elevated GLR was associated with coronary artery disease, acute coronary syndrome, stroke, and composite cardiovascular events [37]. However, most of these studies were cross-sectional or were performed in selected clinical populations of patients with cardiovascular disease. Besides observational studies, various clinical trials have targeted the immune system to reduce the risk of ASCVD. One of the largest is the CANTOS (Canakinumab Anti-inflammatory Thrombosis Outcome Study) randomized trial, in which patients with elevated hs-CRP and previous MI were assigned to receive canakinumab or placebo. Canakinumab selectively inhibits interleukin-1β, resulting in an inhibition of the innate immunity. The 150-mg dose resulted in a lower incidence of recurrent cardiovascular events, compared to placebo [38]. In addition to interleukin-1β, also hs-CRP, fibrinogen, and interleukin-6, all markers in the same cascade, have been shown to have an association with CHD [39–41]. Our findings also show an important role of innate immunity in first-ever stroke and CHD risk in the general population, independent of hs-CRP level or lipid-lowering medication use.

Interestingly, a more recent trial assessing low-dose methotrexate to reduce inflammation in order to prevent recurrent cardiovascular disease showed that methotrexate did not lower levels of interleukin-1β, interleukin-6, or hs-CRP, and that it did not result in a lower number of cardiovascular events compared to placebo [8]. Together with the results from the CANTOS trial, these studies indicate that reducing the risk of cardiovascular events through inflammation may depend on the pathway targeted. In addition to these established pathways, our results indicate that the pathway of adaptive immunity may also affect the risk of cardiovascular disease. More specifically, lymphocyte count, an important player in adaptive immunity, showed protective effects on the risk of ASCVD, although this finding was not statistically significant. Indeed, mobilizing anti-inflammatory T cells to atherosclerotic sites may provide protective responses as has been shown in neurodegenerative disorders [9] and also in cardiovascular disease in experimental and clinical studies [11]. These cells thus additionally serve as a promising therapeutic target to control the proinflammatory processes of atherosclerosis. The significant association we found between GLR and SII and incident ASCVD, although weaker than with granulocytes, supports the hypothesis that ASCVD results from an imbalance between innate and adaptive immunity. Future studies may focus more specifically on targeting both innate and adaptive immunity, rather than only innate immunity, and refine insights into these pathways.

Although previous studies have established the inflammatory hypothesis of atherothrombosis in coronary artery disease, fewer studies have specifically investigated stroke. However, increasing evidence from experimental studies suggests an equally important role for the immune response in stroke [42]. Similarly, we demonstrated that higher levels of granulocytes were associated with stroke of any type, as well as with ischemic and hemorrhagic stroke separately. Especially the latter finding warrants further research since most studies so far have mainly focused on inflammatory responses after the occurrence of an intracerebral hemorrhage, and not before [43].

An interesting finding of our study is that the association between higher levels of granulocytes and risk of ASCVD was more pronounced in younger individuals than older individuals (age below the median [64 years]: adjusted HR [95% CI] for doubled granulocyte count = 2.32 [1.20–4.47], $P < 0.001$; age at or above the median: 1.67 [1.21–2.29], $P = 0.002$). Recently, an

English study and a Dutch study found that while there has been a decline in mortality from acute stroke within 30 days, stroke event rates have increased, and 15-year mortality risk in young adults who have had a stroke remained elevated [44,45]. This suggests that stroke awareness and stroke prevention need to start at much lower age to reduce the occurrence of stroke in younger people. Our results show that inflammation reduction could be a potential important strategy to achieve this.

Moreover, our study showed that coronary and intracranial carotid artery calcifications substantially mediate the association of granulocytes with CHD and stroke, respectively. The observed proportion of mediation for CHD was 19.0%, while for stroke the proportion mediated was 14.9%. This is meaningful given the multiple mechanisms through which innate immunity affects cardiovascular health, including atrial fibrillation (AF) and coagulation [46]. Indeed, the infiltration of immune cells and proteins that cause an inflammatory response in cardiac tissue and circulatory processes is associated with AF [47], and AF is in turn an important risk factor for CHD and stroke. Another possible mediator is blood coagulation, since inflammation results in activation of coagulation, due to tissue-factor-mediated thrombin generation, downregulation of physiological anticoagulant mechanisms, and inhibition of fibrinolysis, increasing the risk of CHD and stroke [48]. Furthermore, besides calcifications, plaques may consist of intraplaque hemorrhage and a lipid core. Although calcifications are an adequate quantitative measure for atherosclerosis, vulnerable plaques without calcification, which are more strongly associated with ASCVD [49], are not quantified by this method. Therefore, it is plausible that the true proportion of inflammation mediated through atherosclerosis of all plaque components is actually higher than the measured proportion mediated through the calcification component.

Blood cell count measurements display short-term variability or measurement error, but we do not expect that this variability affected our results for several reasons. First, we expect to have captured a more chronic or stable effect of immune status because we used blood cell counts measured at multiple time points per participant. Although this longitudinal information was collected intermittently and with error, we could successfully reconstruct the complete longitudinal history by postulating a suitable mixed effects model to describe participant-specific time evolutions. The mixed model accounted for the measurement error problem by postulating that the observed level of the blood cell count measurements equaled the true level of the blood cell count measurements plus a random error term. Thus, the joint models we used accounted for measurement error or variability of the blood cell count measurements. Second, as a sensitivity analysis, we additionally excluded participants with a history of cancer or a history of immune-modulating medication at baseline, and we performed additional adjustments for NSAID use. We performed these sensitivity analyses specifically to address the issue of participants having significantly altered blood cell counts due to underlying disease processes. The results of these analyses did not show significant changes in our effect estimates. Lastly, by extensively correcting for potential confounders such as BMI, we attempted to rule out effects of determinants of blood cell counts as much as possible.

Our study has some limitations. First, studying the etiology of ASCVD, which is the focus of this study, remains difficult within an epidemiological setting due to the limited availability of surrogates for innate and especially adaptive immunity. Second, we could not directly relate the effect of hs-CRP with the different blood cell measurements and ASCVD risk since we did not have hs-CRP measurements available at study baseline, but only 1 round before. Third, the vast majority of our population is of European ancestry (97.6%), potentially limiting generalizability to other ethnicities. Fourth, it is important to note the possible influence of unmeasured confounders on our results. Fifth, due to the cross-sectional nature of the analyses regarding the association between immunity markers and calcifications, there is potential for reverse causality. However, according to the present evidence, it is very plausible that inflammation

comes first since activation of endothelial cells results in increased expression of leukocyte adhesion molecules and the release of chemokines, which attract leukocytes that subsequently infiltrate the vascular wall and drive atherogenesis [50]. Finally, the measurement error for both subclinical ASCVD and immune cell subsets limits the mediation analysis in its reliability. However, since this measurement error introduces random error into the direct effect, it is expected to lead to underestimation of the mediation. Of course, we cannot rule out that it could bias the results in the opposite direction and inflate mediation estimates.

In conclusion, our study shows that increased granulocyte count, GLR, and SII are associated with a higher risk of ASCVD, whereas increased lymphocyte count is protective for ASCVD in the general population. Moreover, higher levels of granulocytes are associated with subclinical atherosclerosis, as measured by arterial calcifications. These calcifications explain a substantial proportion of the link between granulocytes and ASCVD. Our findings support the hypothesis that ASCVD results from an imbalance between innate and adaptive immunity.

## Supporting information

**S1 STROBE Checklist.**
(DOC)

**S1 Analysis Plan.**
(DOCX)

**S1 Fig. ASCVD-free survival and mean change of blood cell count over time in the total population.** The left axis in each panel denotes the blood cell count (dotted line), the right axis the ASCVD-free survival (solid line). All curves were adjusted for all covariates in model II.
(TIF)

**S2 Fig. ASCVD-free survival and mean change of granulocyte count over time stratified by younger and older participants.** Younger participants are those below the median age of 64 years. Of these participants, 142 developed ASCVD, out of 3,799 persons at risk.
(TIF)

**S1 Table. Models for the stratified analyses.** Adjusted for age, sex, education, smoking status, body mass index, diabetes mellitus, systolic blood pressure, diastolic blood pressure, antihypertensive medication, HDL cholesterol, total cholesterol, and lipid-lowering medication. [a]All markers were natural log-transformed (Ln[immunity components $\times 10^3/\mu$l]). *Analysis for each blood cell type adjusted for the baseline blood cell counts of the remaining 2 blood cell types. CI, confidence interval; GLR, granulocyte-to-lymphocyte ratio; PLR, platelet-to-lymphocyte ratio; SII, systemic immune-inflammation index.
(DOCX)

## Acknowledgments

We gratefully acknowledge the study participants of the Ommoord district and their general practitioners and pharmacists for their devotion in contributing to the Rotterdam Study. We also thank all staff that facilitated assessment of participants in the Rotterdam Study throughout the years. Special thanks to Jolande Verkroost–van Heemst for coordinating the follow-up data.

## Author Contributions

**Conceptualization:** Lana Fani, Kimberly D. van der Willik, Maarten J. G. Leening, Bruno H. C. Stricker, Maryam Kavousi, M. Arfan Ikram, M. Kamran Ikram.

**Data curation:** Peter J. Koudstaal.

**Formal analysis:** Lana Fani.

**Methodology:** Lana Fani, Kimberly D. van der Willik, Daniel Bos, Dimitris Rizopoulos, Rikje Ruiter, M. Arfan Ikram, M. Kamran Ikram.

**Software:** Dimitris Rizopoulos.

**Supervision:** M. Arfan Ikram, M. Kamran Ikram.

**Writing – original draft:** Lana Fani.

**Writing – review & editing:** Kimberly D. van der Willik, Daniel Bos, Maarten J. G. Leening, Peter J. Koudstaal, Dimitris Rizopoulos, Rikje Ruiter, Bruno H. C. Stricker, Maryam Kavousi, M. Arfan Ikram, M. Kamran Ikram.

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
