## [Decision Letter · Decision Letter 0]

5 Feb 2020

Dear Dr. Fani,

Thank you very much for submitting your manuscript "Innate and adaptive immunity, subclinical atherosclerosis, and the risk of cardiovascular disease: A prospective cohort study" (PMEDICINE-D-19-03937) for consideration at PLOS Medicine. 

Your paper was evaluated by an academic editor with relevant expertise and sent to independent reviewers, including a statistical reviewer. The reviews are appended at the bottom of this email and any accompanying reviewer attachments can be seen via the link below:

[LINK]

In light of these reviews, we will not be able to accept the manuscript for publication in the journal in its current form, but we would like to invite you to submit a revised version that fully addresses the reviewers' and editors' comments. You will appreciate that we cannot make a decision about publication until we have seen the revised manuscript and your response, and we expect to seek re-review by one or more of the reviewers. 

We hope to receive your revised manuscript by Feb 26 2020 11:59PM. Please email us (plosmedicine@plos.org) if you have any questions or concerns.

Please let me know if you have any questions. Otherwise, we look forward to receiving your revised manuscript in due course. 

Sincerely,

Richard Turner PhD, for Louise Gaynor-Brook, MBBS PhD

Associate Editor, PLOS Medicine

rturner@plos.org

Please include line numbers in your revision.

We suspect that your study reports a retrospective analysis of a prospectively gathered dataset. Therefore, please remove the word "prospective" from your title; we suggest as an alternative "Innate and adaptive immunity, subclinical atherosclerosis and risk of cardiovascular disease: a cohort study."

At the start of your abstract, please add an introductory sentence, say, to provide some background as to why the study was conceived. 

Please combine the "methods" and "findings" subsection of your abstract. The final sentence of the new combined subsection should summarize the study's main limitations. 

Early in the "methods and findings" subsection of your abstract, please use the passive voice (e.g., "... granulocyte and lymphocyte counts were determined ..." [stating interval of data collection]). 

To commence the "conclusions" subsection of your abstract, please begin the sentence "In this study, we found that an increased granulocyte count was associated with a higher risk of ...", or similar. 

Noting the observational design of your study, please avoid language which implies causality (e.g., in the abstract, "The effect of granulocytes on incident CHD ..."). 

In the abstract and elsewhere, please add p values alongside 95% CI where available. 

After your abstract, please add a new and accessible "author summary" section in non-identical prose. You may find it helpful to consult one or two recent research articles published in PLOS Medicine to get a sense of the preferred style. 

Early in the methods section, please state whether your study had a protocol or prespecified analysis plan, and if so attach the relevant document(s) as a supplementary file (referred to in the text). Please highlight analyses that were not prespecified. 

In the paragraph discussing study limitations in the discussion section of your main text, we suggest noting the possible influence of unmeasured confounders. 

Throughout the text, please format reference call-outs as follows: "... neurodegenerative disorders [9,10], but also ...".

Please remove trade marks from the paper. 

Please revisit your reference list to ensure that all reference citations match journal style. Italics should be converted to plain text; where appropriate, 6 author names should be listed, followed by "et al."; please remove "and" from lists of author names; and please ensure that all references include full access details (e.g., references 8, 14, and 34).

Please attach a completed checklist for the most appropriate reporting guideline, which we suspect will be STROBE, as a supplementary document (referred to in the methods section). In the checklist, individual items should be referred to by section (e.g., "Methods") and paragraph number rather than by page or line numbers, as the latter generally change in the event of publication. 

Comments from the reviewers:

*** Reviewer #1: 

[See attachment]

Michael Dewey

*** Reviewer #2: 

I read with great interest the manuscript by Fani et al regarding associations of innate and adaptive immunity with incident myocardial infarction and stroke. The methods appear to be sound and the manuscript is generally well-written. I have a few major concerns/questions as well as a few minor questions. 

The mediation analysis presents large effect sizes, but they are not statistically significant. Based on the confidence interval, they do not appear to be marginally significant. What is your justification for making a strong claim in the conclusion about a mediating effect when you lack such statistical precision? The assessed association between immunity and calcification is cross-sectional and subject to reverse causality. How do we know it's not the calcification that leads to higher granulocytes? At a minimum, the potential for reverse causality should be mentioned in the limitations section of the discussion. In the conclusion of the abstract, "Arterial calcifications explain a substantial proportion" may be overstated given the lack of statistical significance and cross-sectional nature of the immunity and calcification association. 

I would like to see more discussion on the variability and determinants in the measurement of granulocytes, lymphocytes and platelets. How much does any individuals cell count vary from day to day or year to year? I know acute and chronic infections can lead to significant changes in blood cell counts. Could you potentially be measuring an underlying infectious process that may trigger an ASCVD event? Did you consider a sensitivity analysis excluding ASCVD events within the first year of follow up? 

In the analysis presented in table 3, did you consider a sensitivity analysis looking at markers and continuous arterial calcification in those only with non-zero arterial calcification? The magnitude of 0 to 1 for volume may not be same as other incremental increases in arterial calcification.

For the interaction analyses, I would like to see the effect size (confidence interval) for those older than age 64. Given the very low p-value chosen, I would be curious to see the effect estimates for all the interaction analyses tested. There could be meaningful information in these analyses that would warrant at least a place in the supplement section. Given such a profound effect modification by age, I would like the authors to address/hypothesize at least briefly in the discussion how they interpret such findings. 

Methods:

Arterial calcification- I would like to see a little more description of how calcified volume was measured instead of having to refer to multiple previous publications. For instances, did you use a summation of pixels? Were there differences in CT scanners used? 

What is the definition of an immune-modulating medication?

Did you consider a test for non-linearity? The results may be driven by the extremes of the distribution. Consider showing a figure of the dose-response association of blood cell counts with incident ASCVD. 

Did you test for the proportional hazard assumption for Cox modeling?

What is the rationale for using the Bonferroni correction in the interaction analysis, but not your main analysis? If I am not mistaken, typically the p-value for exploratory tests of multiplicative interaction are less stringent than the main analyses. Using such a low p-value for interaction testing may miss important effect modification. 

Minor comments/suggestions:

Abstract:

-In results section the last sentence I believe should have a ",respectively" at the end. 

-Please report the confidence intervals for the mediation analysis effect sizes. 

Methods:

Statistical analysis- "Results can be interpreted as: A doubling of blood cell counts over time in a person compared to another person gives a HR for the risk of ASCVD." Is this a doubling of log-transformed cell counts? If so, this is a different scale and should be specified in the text and tables. For the linear regression analysis, you report a per standard deviation increase while in the cox models you report a per doubling. Was there a reason to use standardized coefficients for one and a doubling for the other?

Paragraph 2, page 13- unclear if the results for blood cells and calcification volume are adjusted for anything. 

Table 1- include abbreviations for all variables (BMI, GLR, PLR, SII, etc.)

Table 2- in caption it says "n= number of incident stroke events", but 3 outcomes are presented above. What are the units for the HR? Is it per standard deviation, per doubling? 

*** Reviewer #3: 

This is a nice paper on the links between the immune system and cardiovascular disease. The methods are pretty advanced, possibly more than absolutely necessary for valid estimates, and are carefully implemented. I do have a few concerns.

The biggest one is the mediation analysis. Mediation analysis has become popular but I think that it has some important limitations in this context. There is measurement error for both subclinical disease and immune cell subsets, both from the technique and the single time point (blood draws are done at a single time and some participants will have abnormal values at this time). Often this will weaken the associations between the three pieces of the mediation analysis (immune cells, subclinical disease, and CVD outcomes). Since random error is put into the direct effect, it has a tendency to underestimate the mediation. Of course, measurement error is complicated, and it could also go in the opposite direction and inflate mediation estimates. At the least this needs to be discussed in the context of interpreting these results in the discussion.

The use of Joint models is nice to see. But I think that what the authors are doing is using a linear mixed model to estimate the change in immune cells over time, assuming a linear change over the time period (which seems to be 6 to 10 years given that visits are 3 to 5 years apart and some small number of participants have 3 blood draws). This isn't a bad model and it will reduce measurement error quite a bit, but it still has pretty strong parametric assumptions that would make sense to explore. Can the authors provide any validation for the intuition of a linear model makes sense? What about using age as a proxy for time, and see if linear models work for the 3 to 5 year age bands? I know that with most participants having only two measurements options are limited, but when using such a cool and cutting edge technique it seems like a good plan to over-explain the assumptions, as readers will be less familiar with them.

The results themselves I have less to say about. Rotterdam is a famously high-quality cohort study with extremely well measured outcomes (and the definitions used were pretty typical). The measures of subclinical disease are well validated and high quality. The blood assays were done by a competent approach. I would have liked further sub-typing of the immune measures, but the world is filled with questions where I wish there was more data. The effect estimates are compatible with existing literature and well reported. But high quality estimates like this are pretty novel. So I am generally pretty happy with the guts of the study. 

Overall, I think a careful writing of the methods section to highlight the assumptions of the advanced approaches used would greatly enhance the interpretability of this manuscript for the target audience. If the authors are convinced that this would disturb the flow of the paper, at the very least a methods online supplement would be appropriate.

***

[LINK]

---

## [Decision Letter · Decision Letter 1]

26 Mar 2020

Dear Dr. Fani,

Thank you very much for re-submitting your manuscript "Innate and adaptive immunity, subclinical atherosclerosis and risk of cardiovascular disease: A cohort study" (PMEDICINE-D-19-03937R1) for review by PLOS Medicine.

I have discussed the paper with my colleagues and the academic editor and it was also seen again by the previous reviewers. I am pleased to say that provided the remaining editorial and production issues are dealt with we are planning to accept the paper for publication in the journal.

[LINK]

We look forward to receiving the revised manuscript by Apr 02 2020 11:59PM. 

Sincerely,

Clare Stone, PhD

Managing Editor 

PLOS Medicine

plosmedicine.org

Requests from Editors:

Please add Rotterdam or Netherlands to the title

Abstract – please be more specific in dates rather than [2002-2015]; also how often is ‘repeatedly’; please add summary demographic information (as well as the stated mean age). 

“while lymphocytes decreased ASCVD risk (HR[95%CI]: 0.87 [0.71–1.06]; P=.17, although not statistically significant” please remove this as it isn’t significant so it’s not appropriate to say it reduces risk. Similarly line 338 – remove. 

Line 74 (author summary), please say ‘may be’ instead of ‘are’

Comments from Reviewers:

Reviewer #1: The authors have addressed all my points.

Michael Dewey

Reviewer #2: The reviewers have more than adequately addressed my comments/concerns. 

Reviewer #3: I think the authors adequately addressed my comments. They engaged in improved language for both mediation and the joint models. The additional analysis run using age was quite interesting and actually improves the evidence base of the paper. A couple of the other reviewer comments seemed important, but I will let them address the adequacy of the response there.

[LINK]

---

## [Editor Report · Decision Letter 2]

10 Apr 2020

Dear Dr. Fani, 

On behalf of my colleagues and the academic editor, Dr. Kazem Rahimi, I am delighted to inform you that your manuscript entitled "Innate and adaptive immunity, subclinical atherosclerosis and association with cardiovascular disease in the Rotterdam Study: A prospective cohort study" (PMEDICINE-D-19-03937R2) has been accepted for publication in PLOS Medicine. 

PRODUCTION PROCESS

PRESS

PROFILE INFORMATION

Thank you again for submitting the manuscript to PLOS Medicine. We look forward to publishing it. 

Best wishes, 

Clare Stone, PhD

Managing Editor 

PLOS Medicine

plosmedicine.org